# Astral hydrogels mimic tissue mechanics by aster-aster interpenetration

Qingqiao Xie[1,2], Yuandi Zhuang[3], Gaojun Ye[3], Tiankuo Wang[4], Yi Cao [4] & Lingxiang Jiang [1,2✉]

Many soft tissues are compression-stiffening and extension-softening in response to axial strains, but common hydrogels are either inert (for ideal chains) or tissue-opposite (for semiflexible polymers). Herein, we report a class of astral hydrogels that are structurally distinct from tissues but mechanically tissue-like. Specifically, hierarchical self-assembly of amphiphilic gemini molecules produces radial asters with a common core and divergently growing, semiflexible ribbons; adjacent asters moderately interpenetrate each other via interlacement of their peripheral ribbons to form a gel network. Resembling tissues, the astral gels stiffen in compression and soften in extension with all the experimental data across different gel compositions collapsing onto a single master curve. We put forward a minimal model to reproduce the master curve quantitatively, underlying the determinant role of aster-aster interpenetration. Compression significantly expands the interpenetration region, during which the number of effective crosslinks is increased and the network strengthened, while extension does the opposite. Looking forward, we expect this unique mechanism of interpenetration to provide a fresh perspective for designing and constructing mechanically tissue-like materials.

[1] School of Molecular Science and Engineering, South China University of Technology, Guangzhou, China. [2] South China Advanced Institute for Soft Matter Science and Technology (AISMST), South China University of Technology, Guangzhou, China. [3] College of Chemistry and Materials Science, Jinan University, Guangzhou, China. [4] Collaborative Innovation Center of Advanced Microstructures, National Laboratory of Solid State Microstructure, Department of Physics, Nanjing University, Nanjing, China. ✉email: jianglx@scut.edu.cn

An ongoing endeavor in chemistry and materials science is to develop synthetic matter that recapitulates mechanical features of biological cells, tissues, and organs. Significant progress has been achieved on the subcellular and cellular levels[1,2]. Early studies revealed a strong shear-stiffening response universal to cytoskeletal and extracellular biopolymer networks (Fig. 1a, c); this characteristic response was mainly ascribed to the semiflexible nature of biopolymers and it can protect the cells from large, integrity-threatening deformations[3,4]. In the synthetic regime, the shear-stiffening response was neatly reproduced in a number of biomimetic systems such as polyisocyanopeptide[5,6] or

rodlike micelle hydrogels[7–9], signifying the importance of chain stiffness and crosslinking strength.

The paradigm is currently shifting towards higher complexity, in particular, from uniaxial to multiaxial mechanics[10] and from cellular to tissue levels[11–13]. Traditional shear experiments are uniaxial in nature with the applied/measured stress or strains in a single (linear or circular) direction. This configuration is an oversimplification of the in vivo situations in which cells and tissues are constantly subject to complex, multiaxial mechanical stimuli. A step forward is biaxiality with additional stimuli superimposed on the axial direction orthogonal to the shear plane

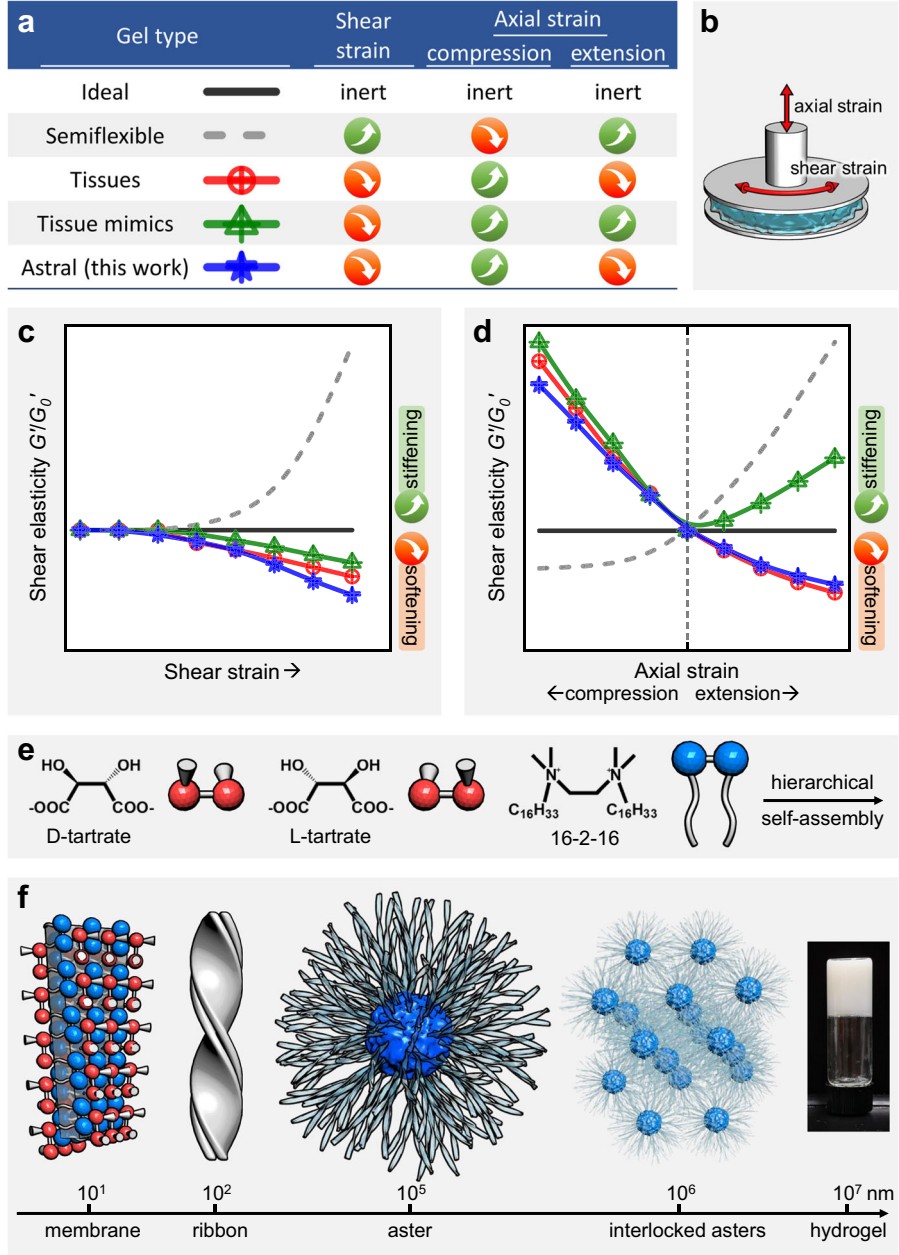

**Fig. 1 Overview of gel mechanics in response to shear and axial strains.** Mechanical responses of five kinds of gels (**a**, **b**) and schematic representations of astral hydrogels (**e**, **f**). **a** A table to summarize the shear and biaxial responses of ideal gels of flexible chains, networks of semiflexible polymers, tissues, hybrid tissue mimics of biopolymer networks with embedded colloids, and astral hydrogels. The tissues are limited to connective and nerve tissues. Notably, semiflexible networks are mechanically tissue-opposite while astral gels are tissue-like. **b** A cartoon denoting shear and axial directions in a typical two-plate rheometer configuration. **c**, **d** Representative curves of normalized shear elasticity as a function of shear strain (**c**) or axial strain (**d**); the absolute magnitudes are not to be taken seriously. **e**, **f** Cationic gemini molecules (16-2-16) with chiral counterions self-assemble into hierarchical asters with an amorphous core and divergently growing, crystalline ribbons. The asters interlock with one another to form an extensive, elastic network.

(Fig. 1b), informing for example the dependence of shear moduli on the axial strain.

Ideal gels made of cross-linked flexible polymers are inert to shear and axial strains before failure (Fig. 1a, c, d)[14]. Networks of semiflexible polymers (biological or synthetic in nature) are usually shear-stiffening, compression-softening, and extension-stiffening[15,16]. Please note that the stiffening or softening is specific to the shear direction throughout this paper. One may intuitively expect this set of mechanical characteristics to prevail at the tissue level because tissues are essentially assemblies of cytoskeletal and extracellular networks. To one's surprise, many connective and nerve tissues respond in the exact opposite ways —moderate shear-softening, remarkable compression-stiffening, and slight extension-softening (Fig. 1a)[17–19]. Janmey et. al. attributed the compression-stiffening response to an interplay between the nonlinear elasticity of biopolymers and the dense inclusion of cells; they also replicated this feature (but not the extension-softening response) in a hybrid system of biopolymers and colloidal inclusions[20]. Although the shear-softening behavior is frequently observed in synthetic hydrogels, tissue's biaxial responses are not yet emulated by a fully synthetic system. Recently, we studied the hierarchical self-assembly of a cationic gemini surfactant (16-2-16) with chiral counterions (a mixture of D- and L-tartrate) into synthetic asters of an amorphous core and radial, crystalline ribbons (Fig. 1e, f)[21]. The asters resemble microtubule asters in terms of structural regularity, elasticity, and positioning capability[22].

In this paper, we report that densely packed asters with moderate interpenetration constitute astral hydrogels featuring tissue-distinct geometry yet tissue-like mechanics (shear-softening, compression-stiffening, and extension-softening). The astral hydrogels, in contrast to common gels, are interpenetrative in nature; moderate aster-aster interpenetration provides mechanical strength and its response to axial strains render the gels mechanically tissue-like. The current mechanism of interpenetration is fundamentally different from the mechanism of cell- or particle-inclusion proposed for tissues or hybrid tissue mimics. We envision the astral gels to serve as an example of interpenetrative matter that could be appealing in geometry and mechanical properties.

## Results

**General aspects of astral hydrogels**. We prepare a series of 16-2-16 with a different enantiomeric excess (ee) of the L- and D-counterions, in which ee is defined as the relative concentration difference $(C_L - C_D)/(C_L + C_D)$, from 0 (racemic) to 1 (pure L-tartrate)[23,24]. We dissolve the surfactant powder in hot water (80 °C) and cool the solutions to room temperature, during which asters appear to cloud and solidify the samples in 2 h. A sample is classified as a gel if it does not flow when the vial is inverted or gently shaken. The critical gel concentration is around 1 wt% basically irrespective of ee from 0 to 0.75. Images from optical microscopy (OM) and scanning electron microscopy (SEM) shows prevailing asters featuring well-defined core-ray structure and large sizes ~50–120 μm (Fig. 2a–e). The radial ribbons are semiflexible fibers with a persistence length similar to that of microtubules[23]. While the architecture of single asters was characterized in detail previously[21], we confirm here that each one is a loose, radial skeleton occupying a spherical volume of ~99% water (Fig. 2a, b). In gel slices, the asters are densely packed to form an extended network that can trap solvent (Fig. 2c–e for ee = 0, 0.33, and 0.5, respectively).

The ee has a dramatic effect on the helicity of aster ribbons (Supplementary Fig. 2 in ref. [22]), but its effect on the overall astral geometry is limited. Visual inspection of the microscopic images

(Fig. 2c–e) suggests that the asters are of similar sizes and similar ribbon densities for different ee values. Notably, the characteristic radial symmetry is shared by astral gels of different compositions but in stark contrast to the random percolation of polymers or fibers in common gels. Closer inspection reveals that the neighboring asters are moderately interpenetrated (Fig. 2c–e). In the intersection region, the peripheral ribbons from two opposing asters can crosslink with one another by interlacement, as highlighted by 2D images (Fig. 2f, g) and by a 3D reconstruction (Fig. 2h). When a gel slice is compressed by the top coverslip, distinctive "walls" are developed between adjacent asters possibly due to intensive interlacement (Fig. 2e).

**Shear and biaxial mechanics of astral hydrogels**. While most in vitro rheological measurements have been conducted in a uniaxial/shear configuration, in vivo cells and tissues are constantly subjected to multiaxial mechanical stimuli. In a typical shear configuration, oscillatory stress (or strain) is applied in a circular direction and a strain (or stress) is then measured in the same direction, giving elastic and viscous moduli ($G'$ and $G''$). For a representative astral gel (ee = 0.33 and C = 1.5 wt%), the frequency-moduli plot indicates that the sample is a strong gel with $G'$ dominating $G''$ in the entire frequency range. A shear strain sweep reveals a shear-softening response with viscous dissipation taking over beyond the yield point ~4%. Similar curves are obtained for other astral gels with elasticity insensitive to ee (Supplementary Fig. 1) and positively dependent on C. Although these shear properties are similar to those of tissues, they are frequently observed in many synthetic hydrogels[25,26].

For biaxial measurements, a gel sample is first formed in situ between two parallel plates and a finite axial strain is then applied by vertically shifting the top plate. The axial strain is usually between −35 to 10%, negative for compression and positive for an extension. With a given axial strain, the normal force is recorded and routine shear measurements are performed to give $G'$ and $G''$. The shear moduli-shear strain curves under different axial strain are represented in a 3D fashion (Fig. 3c), in which the curves are similar in shape but shift upward and rightward upon compression. Highlighted by the droplines to the left and bottom walls, we can notice that the elasticity increases and the yield point moves to larger strains upon compression. $G'$ in the linear regime is extracted and plotted as a function of axial strain (Fig. 3d, e), in which compression-stiffening and extension-softening responses are observed for all the astral gels. In addition, the slope is sharper for higher concentrations (Fig. 3d) but largely invariant to ee (Fig. 3e). This invariance to ee is likely related to the fact that the compression-stiffening mechanics is mostly affected by the overall astral geometry (which is not sensitive to ee as shown in Fig. 2c–e) rather than any molecular specifics or ribbon properties, as we will discuss in the minimal model. These biaxial properties are parallel to those of tissues and are quite unique for this particular class of synthetic hydrogels.

**Microscopic insights into aster-aster interpenetration**. Although optical tweezering could be an ideal means to manipulate the asters and measure the forces simultaneously[27], its trapping force are way too small to affect the asters (~100 μm). We thus employ optical fibers to physically maneuver the asters. When one aster is pushed against another, the formation of an aster-aster intersection is followed in real time (Fig. 4a, b). The process starts by marginal contact of the peripheral ribbons, proceeds by ribbon interdigitation during which interlacement may take place, and ends by forming a dark wall. Further

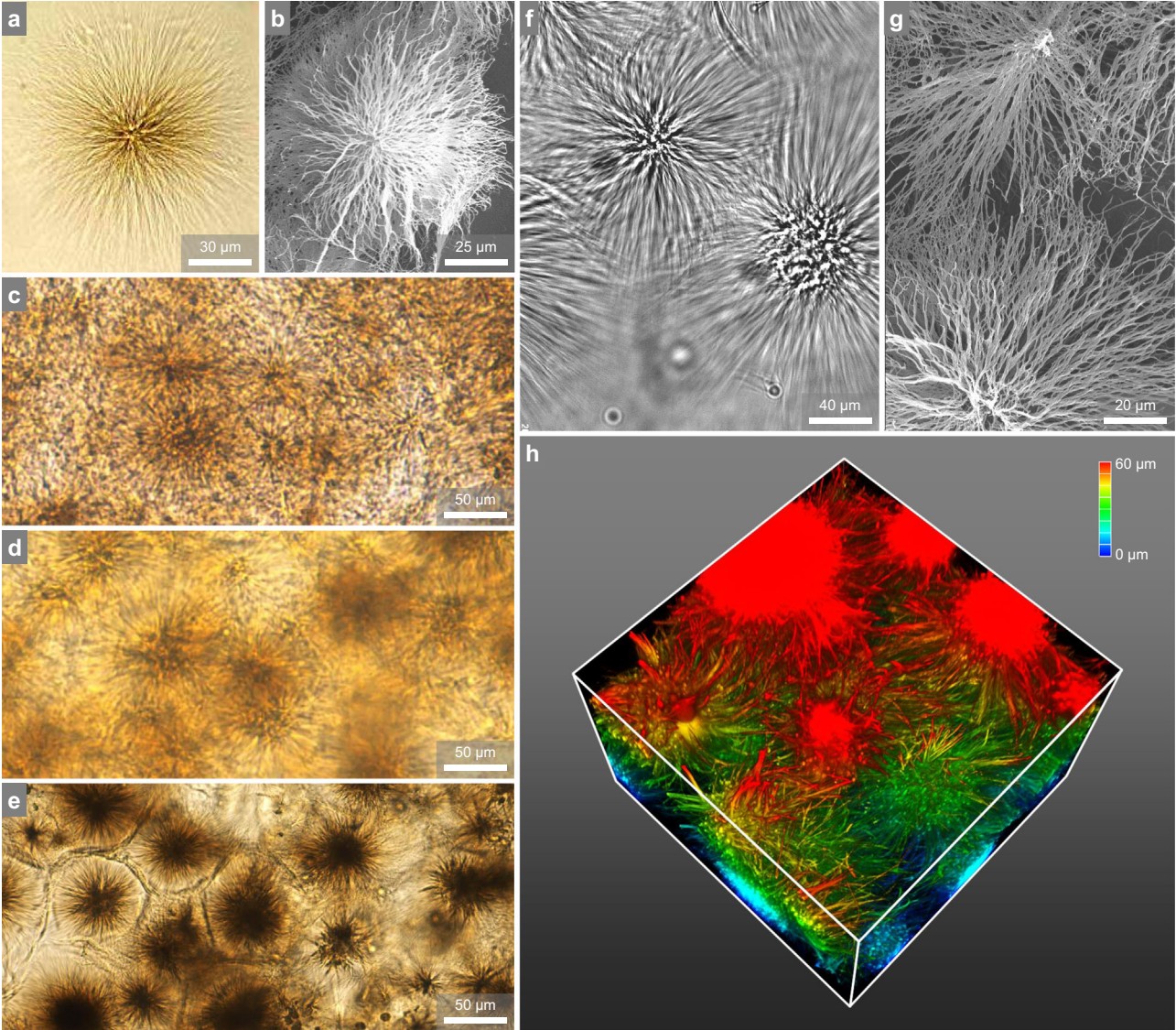

**Fig. 2 Microstructures of asters and astral gels. a, b** Images of single asters by optical microscopy (**a**) and SEM (**b**). **c–e** Densely packed asters in astral gels ($C = 1.5$ wt% and ee = 0, 0.33, and 0.5, respectively). **f, g** Aster-aster interpenetration is highlighted by optical microscopy (**f**), SEM (**g**), and 3D reconstruction of CLSM (**h** box size = $80 \times 80 \times 40\,\mu m$, the color bar indicates the height in the $z$-direction).

manipulations highlight the mechanical function of the aster-aster interpenetration—they provide significant elasticity against stretching or compression parallel to the aster-aster connection and, notably, against shearing perpendicular to the aster-aster connection. The aster-aster junction will remain intact unless the two asters are forcefully pulled away by an optical fiber.

To study the effect of compression semiquantitatively, we design the following experiments (schematically illustrated in Fig. 4c) to sidestep the incapability of force detection by optical fibers. A small piece of astral gels is placed above a glass slide and a cluster of natively interpenetrated asters near the boundary of the piece are clearly visible. The average distance between neighboring aster centers is recorded as $D_{00}$. An optical fiber is used to shear this cluster of asters by moving the outmost aster parallel to the boundary until aster-aster detachment is observed ($D_{10}$). The ratio $D_{10}/D_{00}$ reflects maximal shear strain before failure in the native state, roughly comparable to the global yield point measured by rheology. Another cluster of asters is first compressed to maximum ($D_{01}$) and then sheared until failure ($D_{11}$), so the ratio $D_{11}/D_{10}$ reflects the microscopic

yield point in a pre-compressed state (Supplementary Fig. 2 and Supplementary Movie 1). The experiments are repeated five times for different pieces of astral gels and averaged data are reported in Fig. 4d, where $D_{11}/D_{10}$ is clearly larger than $D_{10}/D_{00}$. More microscopic results on samples with different ee values are displayed in Supplementary Fig. 3 and Supplementary Movies 2, 3. This result suggests that the aster clusters can sustain a greater strain in the pre-compressed state than in the native state, in broad agreement with the macroscopic results of yield point shift (Fig. 4e).

These microscopic results motivate us to argue that the moderate aster interpenetration in the native state can mechanically support the gel and that axial strain can advance or withdraw such interpenetration, effectively stiffening or softening the gel. These arguments are further substantiated in the following section.

**A minimal model to describe the axial mechanics of astral gels.** We plot the normalized elasticity $G'/G'_0$ of astral gels as a

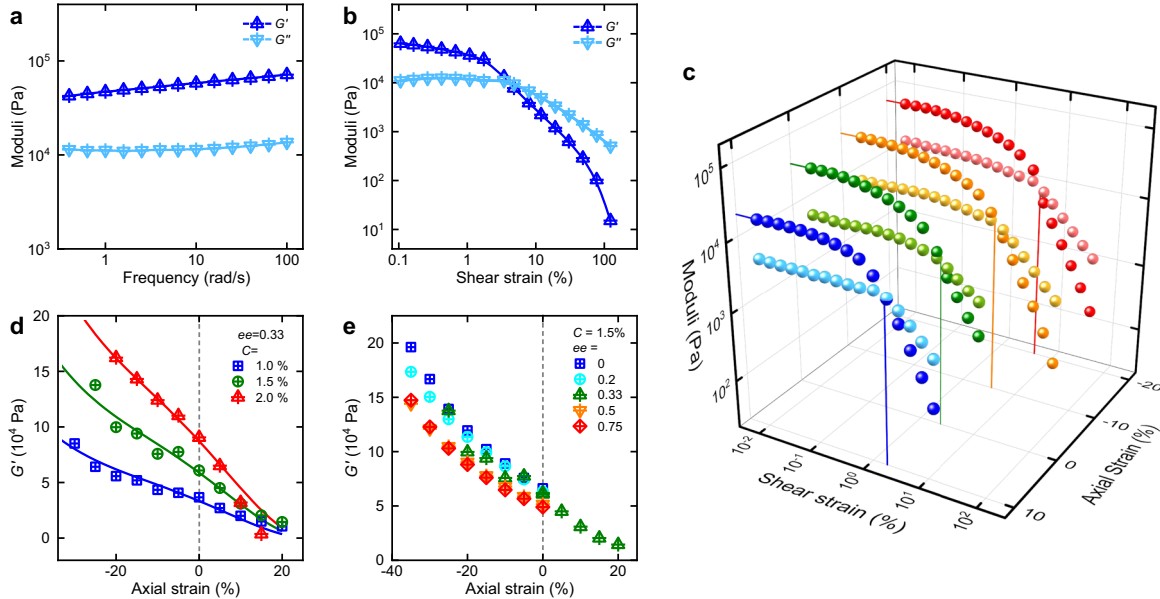

**Fig. 3 Shear and biaxial responses of astral hydrogels. a**, **b** Routine shear rheology at zero axial strain for a representative astral gel ($C = 1.5$ wt% and ee $= 0.33$) with a frequency sweep (**a** shear strain fixed at 0.1%) and shear strain sweep (**b** frequency fixed at 10 rad/s). **c** A 3D plot of moduli-shear strain curves under different axial strains (gel composition: $C = 1$ wt% and ee $= 0.33$). The dark and light color spheres denote $G'$ and $G''$; droplines to the left wall highlight elasticity in the linear regime and droplines to the bottom wall indicate yield points. **d**, **e** Dependence of shear elasticity on axial strains for different concentrations (**d** solid lines are obtained by vertically rescaling the global fitting in Fig. 5a) and various ee (**e**).

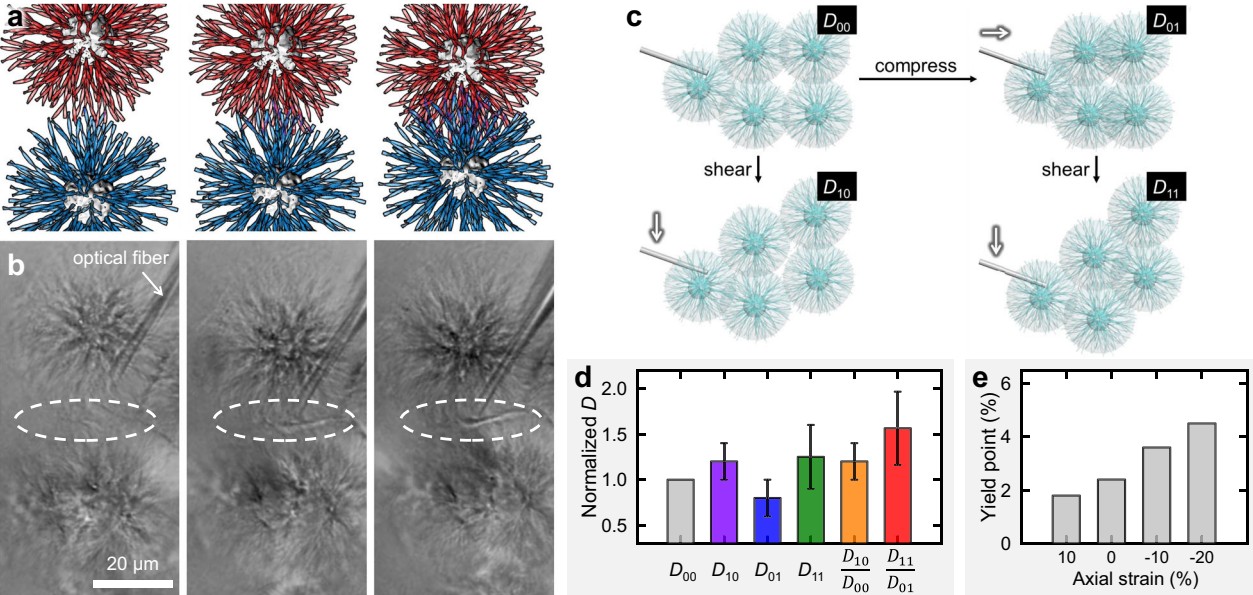

**Fig. 4 Microscopic insights into aster-aster interpenetration. a**, **b** Formation of an intersection is followed in real time by pushing one aster against another with an optical fiber. **c** A schematic diagram showing the protocol to deform a cluster of asters. In our denotations, $D$ represents the average distance between neighboring aster centers, while the first subscript indicates unsheared (0) or sheared (1) and the second subscript uncompressed (0) or compressed (1). **d** A bar diagram showing measured $D$ values averaged over measurements on five different clusters. Error bars represent standard deviations ($n = 5$). **e** Yield point as a function of axial strain (data taken from Fig. 3c).

function of stretch ratio $\lambda$ (<1 for compression and >1 for extension) in Fig. 5a. This is the key figure of this work in which all the data points across different gel compositions collapse into a single master curve. This data similarity prompts us to develop a minimal model to capture the structural essences of astral gels and to account for the observed biaxial mechanics. Among different factors, of particular interest here are the number of crosslinks $n_c$, sample volume $v_T$, and local filament modulus $K$,

which are related to shear elasticity by[14]

$$G' \propto \frac{1}{v_T} n_c K \qquad (1)$$

While all three terms are inert to axial strains for ideal gels, they are highly dependent on $\lambda$ for astral gels. First, the sample volume is not conserved given the observation of water outflux/influx during compression/extension. Instead, the lateral dimensions are

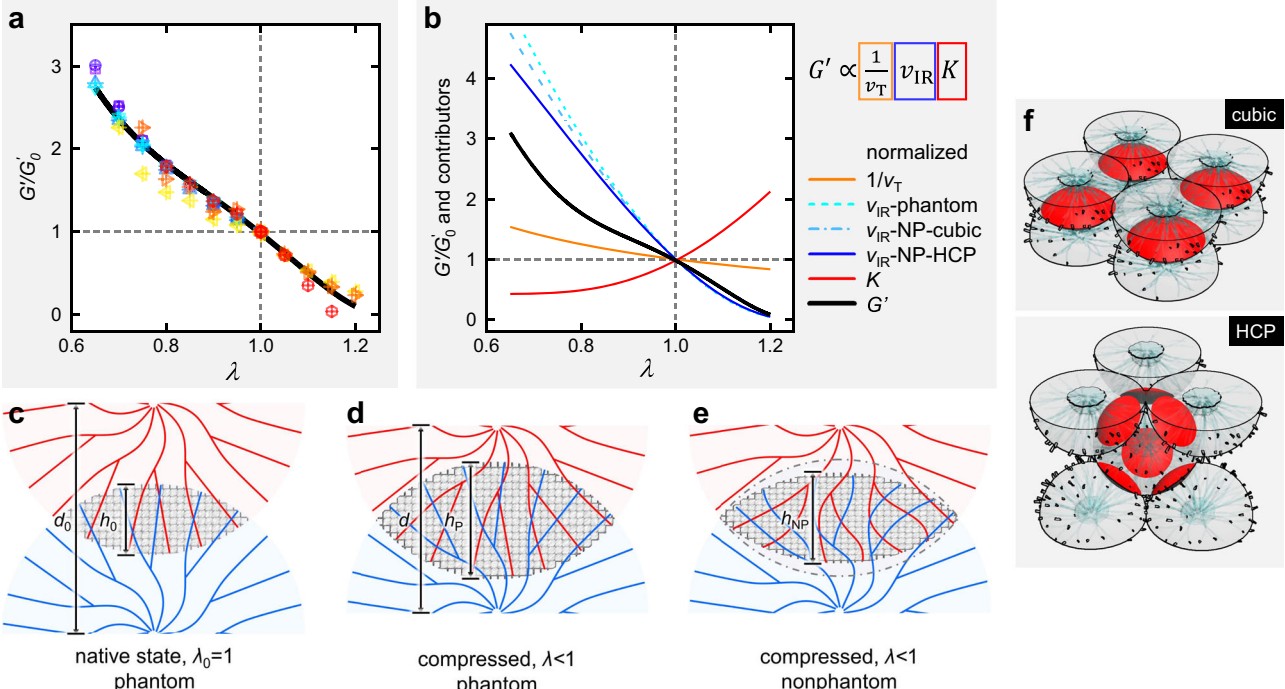

**Fig. 5 A master curve and a minimal model. a** Normalized shear elasticity $G'/G'_0$ of astral gels as a function of stretch ratio $\lambda$, where the experimental data points from Fig. 3d, e collapse to a single master curve. The black line is a global fitting with $d_0 = 0.79R$ and $T = 0.15$. **b** Dependence of normalized elasticity and its contributors on stretch ratio, as predicted by the proposed minimal model with input parameters $d_0 = 0.8R$ and $T = 0.2$. **c–e** Schematic sketches of two axially stacked asters under compression, in which the interpenetration regions in phantom or nonphantom states are highlighted by gray grids. Note that the interlaced ribbons are undisturbed in phantom limit (**c**, **d**) but buckled in the nonphantom condition (**e**). **f** Cubic packing and hexagonal close packing (HCP) of monodispersed asters, where the interpenetration regions are stressed in red.

conserved by the top disk such that the volume scales linearly with $\lambda$, reading[28]

$$v_T \propto \lambda \qquad (2)$$

This term corresponds to network densification/dilution upon compression/extension.

Second, we recall that the ribbon interlacement in the interpenetration region can act as effective crosslinks to sustain stress or strain, then $n_c$ is proportional to the interpenetration volume $v_{IR}$

$$n_c \propto v_{IR} \qquad (3)$$

provided that the number density of ribbons is roughly constant in an aster. The full derivation of $v_{IR}$ is presented in the Methods section, while the essentials are sketched here. Consider two identical asters (diameter $= R$) moderately interpenetrated in the axial direction with a center-to-center distance $d = d_0$ (Fig. 5c). In the limit of phantom interpenetration (ribbons from one aster can freely pass their counterparts from the other aster), the spherical shape of the two asters is perfectly retained and interpenetration is thus maximized (with a height of $h_P$, Fig. 5d). However, the preexisting interlacement will resist further interpenetration (or depenetration) to a certain degree upon compression (or extension), deforming the interpenetration region to $h_{NP} < h_P$ as shown in Fig. 5e (or $h_{NP} > h_P$). We describe the local deformation by the global strain $\Delta\lambda$ and transmittance $T$ (denoting the fraction of deformation transmitted from the globe to the interpenetration region). Eventually, $v_{IR}/v_{IR}^0$ is parametrically dependent on the condition of initial interpenetration ($d_0$) and on $T$ (Supplementary Fig. 4). Also, $v_{IR}/v_{IR}^0$ is insensitive (Fig. 5b) to specific aster arrangement such as cubic or hexagonal close packing (HCP, Fig. 5f).

Third, it was well studied that the effective filament modulus $K$ of semiflexible polymers is affected by axial strains[15,16]. Specifically, compression introduces filament buckling and stronger undulations, both of which weaken the network; extension stretches filaments towards their full length and suppresses undulations, enhancing the network. Experimental, simulation, and theoretical results revealed that these responses are universal for semiflexible polymers and insensitive to the filament persistence length or concentration[13,14,28]. We thus seek to adapt the reported dependence to model the effect of intersection deformation on the interlaced ribbons. Since no analytical equation is available for the current compression-extension range, we choose to digitize the simulation curve in ref. [16] by fitting it to a polynomial function $K_{poly}$, obtaining.

$$K \propto K_{poly}(\lambda, T) \qquad (4)$$

Finally, we combine the three terms to get a $G'/G'_0$ curve and dissect their respective contributions (Fig. 5b for $d_0 = 0.8R$ and $T = 0.2$). The network densification/dilution ($v_T$) and aster interpenetration ($v_{IR}$) are in favor of compression-stiffening and extension-softening, while the filament buckling/stretch ($K$) act against this trend. Clearly, aster interpenetration is the dominating term to determine the overall trend for $G'/G'_0$; the other two terms aid to finely shape the curve into a resemblance of the experimental data. Indeed, a two-parameter fitting matches the experimental data satisfactorily (the black line in Fig. 5a), giving $d_0 = 0.79R$, in line with the observed moderate interpenetration in the native state, and $T = 0.15$, suggesting that a small fraction of the global strain is transmitted to the interpenetration region. The collapse of the experimental data reflects that the astral gels of different compositions are of similar $d_0$ and $T$ values. A simple rescaling of this global fitting overlaps with individual $G'$ curves fairly well (lines in Fig. 3d).

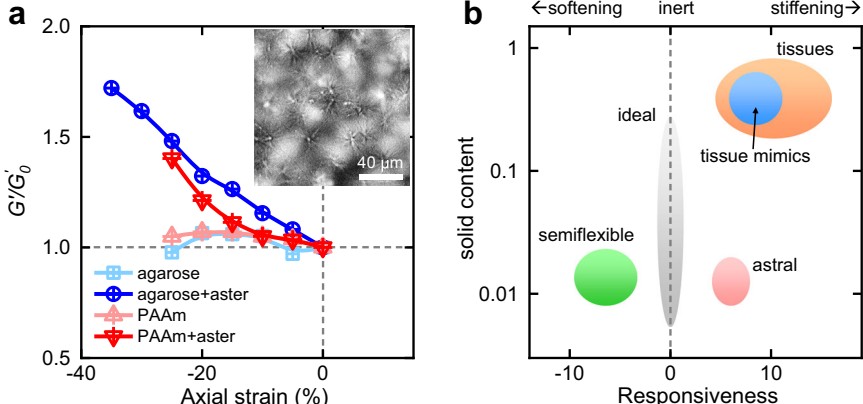

**Fig. 6 Comparison of astral to other gels. a** Introduction of asters turns the inert gels of agarose or PAAM into compression-stiffening. The inset is a picture of asters in agarose gel. **b** A map of responsiveness and solid contents for five kinds of gels. The astral gels are moderately compression-stiffening with the least solid contents (the red area).

**Turning inert gels into compression-stiffening by adding asters**. The abundant free space among the loose skeleton of asters can potentially accommodate other particles or structures towards double networks or composite materials. Along this line of thinking, we load 16-2-16 monomers into precast inert gels of agarose or polyacrylamide (PAAm) via infiltration at an elevated temperature and trigger aster formation by cooling. The asters appear intact albeit thinner in the presence of preexisting networks; the inert response of the agarose or PAAm gels is now altered into moderate compression-stiffening (Fig. 6a). There is a critical concentration of asters (1 to 2 mg/ml) for the PAAm and agarose gels, below which the asters are not populated enough to form a connective network, and above which compression-stiffening is observed and the overall elasticity ($G'$) increases with aster concentration. The slope of compression-stiffening is weakly dependent on aster concentration. We thus expect the asters to act as a promising additive to turn an inert gel into tissue-like if their formation is not hindered.

## Discussion

Compression-stiffening mechanics was observed in many connective and nerve tissues, such as adipose, kidney, liver, lung, brain, and spinal cord tissues[17–19]. It is, however, not clear yet how would epithelial or muscle tissues respond in the shear direction under axial compression. Given the significant difference between different tissue types (for example cells are tightly packed with limited extracellular matrix in epithelial tissues)[29], they may well behave differently. It is therefore to be noted that the compression-stiffening behavior is not necessarily generic for all four tissue types.

Nevertheless, Janmey et. al. recently proposed a model to explain the compression-stiffening behavior in connective and nerve tissues, in which they fabricated hybrid tissue mimics with biopolymer networks and colloidal inclusions[20,30]. They underlay the importance of embedded cells or colloids, which can convert the global compression into the local stretch for the semiflexible filaments adjacent to the cells or colloids. If the inclusions are densely populated, the local effect of stretch-stiffening prevails such that the tissues or mimics behave in a compression-stiffening manner. But this mechanism cannot fully account for the extension-softening behavior of tissues.

It is worth noting that the extracellular matrix and common gels formed by percolated networks of randomly arranged fibers or polymers are generally non-interpenetrative, but the current astral gels are interpenetrative. It is thus surprising to see that the astral gels and tissues are structurally different, yet their

compression-stiffening responses are similar. The current mechanism of aster-aster interpenetration is fundamentally different from the mechanism of cell- or particle-inclusion proposed for tissues and their hybrid mimics; it may provide an alternative yet effective route towards tissue-like mechanics.

In a comparative context, we consider two parameters, responsiveness and solid content for tissues and different types of gels (Fig. 6b). The former is defined as the sharpness of compression-stiffening, mathematically given by the negative slope of $G'/G'_0$-$\lambda$ curves in the range of $\lambda = 0.9$ to 1. The responsiveness is negative for compression-softening gels of semiflexible polymers[15,16] and close to zero for ideal gels as expected. For three compression-stiffening materials, connective and nerve tissues cover a broad range from 4 to 18[17–19], while the hybrid tissue mimics[20] are positioned around 8 and astral gels around 5. Although the astral gel is less pronounced in responsiveness, their solid contents (~0.01) is lower than those of tissues (~0.25 to 0.85) and tissue mimics (~0.5) by one or two orders of magnitudes (Fig. 6b). Therefore, the astral gels can attain reasonable axial responses with minimal solid contents, desirable for massive production and economic applications.

Spatial variation of the gel mechanics is assessed by atomic force microscopy (AFM, see Supplementary Fig. 5 for details). We observed a significant fluctuation in Young's modulus ($E$) from 0.3 to 50 kPa in a $25 \times 20\,\mu m^2$ area, suggesting that inhomogeneity in aster sizes and packing could affect local mechanical properties. But this spatial variation cannot be directly related to compression-stiffening response as the AFM measurements are essentially uniaxial.

One potential application of the astral gels is to host cells as extracellular matrices. A dire issue, however, is the significant cytotoxicity of 16-2-16 because of its dimeric cationic headgroups (see Supplementary Figs. 6 and 7 for details). Although the current astral hydrogels themselves are not suitable for cell culture or encapsulation, we argue that the key finding of this work lies in the interpretative nature of the astral gel beyond its molecular details. If one can replace a 16-2-16 molecule with a cytocompatible molecule that forms asters, it is then possible to make cytocompatible astral gels. Alternatively, it is possible to deposit of a layer of silica or other inert materials onto the surface of asters to screen the 16-2-16 molecules, minimizing its toxicity and enhancing its stability.

In summary, we present here a class of interpenetrative materials, the building block of which is asters with a common core and divergently arranged ribbons. The asters interpenetrate with their neighbors via interlacement of peripheral ribbons to

form a water-trapping, elastic network. The astral gels exhibit compression-stiffening and extension-softening responses in resemblance to tissues with striking regularity—all the experimental data across different gel compositions collapse to a single master curve. A minimal model is constructed to reproduce the master curve in full, dissecting three contributors. The interpenetrative nature plays a central role, in the sense that compression furthers interpenetration to stiffen the network and extension does the opposite, while the nonlinear elasticity of the semiflexible astral ribbons and network densification/dilution are secondary. We envision the current astral gels to set an example for the interpenetrative matter that could be appealing in geometry and mechanical properties.

## Methods

**Materials.** 1-Bromohexadecane (98%) were purchased from Adamas. Silver carbonate (Ag₂CO₃, 99.9%), L-tartaric acid (99%), D-tartaric acid (99%), acrylic acid (AAc, >99.7%), N,N′-methylenebis(acrylamide) (BIS, ≥99.0%), potassium persulfate (KPS, 99.99%), and sodium chloride (≥99.5%) were purchased from Aladdin Chemicals. Deionized water (18.2 MΩ cm) was produced by a Milli-Q water purification system (Milli-pore, USA). The surfactant was synthesized as reported[22]. N,N,N′,N′-tetramethylethylenediamine (0.1 mol) and 1-Bromohexadecane (0.8 equiv.) were heated in MeCN (200 ml) at 40 °C for 1 day. Subsequent evaporation and crystallization from ethanol produce the gemini surfactant ethylene-1,2-bis (cetyldimethylammonium) (16-2-16) with bromide counterions. The counterion exchange from bromide to tartrate was performed at strict stoichiometry by mixing the surfactant with a suspension of the silver salt of tartaric acid in methanol. Stoichiometry and purity of the product were confirmed by element analysis and nuclear magnetic resonance (NMR).

**General methods.** We weighed the desired amount of surfactant powder in water (C = 1 to 2 wt% and ee = 0, 0.33, 0.5, or 0.75), dissolved the powder at 80 °C, and then incubated the sample at 25 °C for at least 4 h for astral networks to be fully developed. The critical temperature of aster formation is around 45 °C, and gel formation is insensitive to cooling rate (controlled cooling from 1 to 10 °C/min or ambient cooling). The gel samples were stable for at least 2 weeks as confirmed by rheology and OM. To load asters into inert gels, we first prepare agarose or PAAm hydrogels by cooling or UV irradiation, respectively. The precast inert gels were immersed in hot 16-2-16 solutions for 4 h allowing infiltration of the monomers and were then cooled to trigger aster formation. The astral gels or aster-loaded gels were observed by an optical microscope (Olympus BX51, ×40 objective), a confocal laser scanning microscope (CLSM, Leica TCS SP8, ×60 oil objective, fluorescently dyed by Nile red), and a SEM (Zeiss EVO MA15, samples freeze-dried).

**Rheology.** Rheological measurements were performed on a rotational rheometer (TA DHR-2 with a flat plate clamp). In oscillatory shear experiments, the strain is fixed at 0.1% and frequency varied (Fig. 3a), or the frequency is fixed at 10 rad/s and strain varied (Fig. 3b). To introduce a finite strain on the axial direction, the top plate was vertically repositioned with a typical stretch ratio between 0.65 and 1.2. For biaxial measurements, a gel sample is first formed in situ between two parallel plates and a finite axial strain is then applied by vertically shifting the top plate. For any given axial strain, the normal force is recorded and routine shear measurements are performed to give G′ and G″ (Fig. 3c). Rheological measurements were repeated three times. Notably, we surround the sample and plate edges with a reservoir of water such that the gel can expel and absorb water during compression and extension, respectively.

**Manipulation by an optical fiber.** The fiber tip was fabricated from a commercial multimode optical fiber by a flame-heating technique. First, the polymer jacket of the fiber was stripped off to obtain a bare fiber of 3.0 cm in length. The bare fiber was heated by the outer flame of an alcohol lamp for 50 s until the fiber reached its melting point. Then, the fiber was drawn at a rate of ~4 mm/s, which gradually tapers off. Finally, the diameter of the fiber tip shrank from 125 to 1.3 μm within an axial distance of 52 μm. This fiber tip was mounted on a tunable fiber micromanipulator (Kohzu Precision Co., Ltd.), allowing us to operate it in 3D at a precision of 50 nm. To manipulate individual asters or small clusters, we cut a small piece of the astral gel and disperse it in excess water such that the otherwise intact and tight block of asters is loosened and scattered. Similar microscopic manipulation is always performed multiple times for different gel compositions to ensure that the observed behavior is generic for the current aster system. We designed the following protocol to deform aster clusters. First, a cluster of natively interpenetrated asters near the boundary of a piece of astral gels is identified ($D_{00}$). An optical fiber is used to shear this cluster of asters by moving the outmost aster parallel to the boundary until aster-aster abjunction is observed ($D_{10}$). Another cluster of asters is first compressed to maximum ($D_{01}$) and then sheared until failure ($D_{11}$). In our denotations, $D$ represents the average distance between

neighboring aster centers, while the first subscript indicates unsheared (0) or sheared (1) and the second subscript uncompressed (0) or compressed (1).

**Atomic force microscopy (AFM).** Spatial variation of Young's modulus ($E$) was mapped out for a layer of astral gel by an AFM instrument (JPK Nanowizard II, JPK Instruments, Berlin, Germany) at room temperature. The common sharp tip was replaced by a 7-μm silica sphere to ensure smooth compression on loose astral ribbons; to do so, the silica sphere was attached to the silicon nitride tip (type MLCT, Bruker Company) by epoxy glue. A cantilever (spring constant in the range of 0.03–0.07 N m⁻¹) was positioned on top of a thin layer of astral gel. Multiple (200 to 300) intention-retraction curves were retrieved for a single position; identical measurements were performed on 20 positions (a 5-μm apart, 5 × 4 lattice). Each approaching curve was fitted by the Hertz model to give $E$ and an averaged value was obtained for different positions on the gel layer.

**Cell viability in the gel extract.** The cell line, American ATCC mouse fibroblast L929, was used in this study. The cell culture medium consisted of 10% bovine serum albumin (Gibco), 1% penicillin-streptomycin (Gibco), and DMEM (Gibco). Cells and sterilized dishes were cultured in 5% CO₂ cell incubator at 37 °C. Cell counting kit 8 (CCK-8, Dojindo) assay was used for cytotoxicity evaluation of extracting solutions from the astral gels. First, L929 cells (5 × 10³ cells/mL) in the growth phase were inoculated in a 96-well plate (100 μL/well), and then the cells were cultured in a mixed solution of extract and culture medium. The volume fraction of the extract was 0, 12.5, 25, 50, and 100%, among which 0% was taken as the blank group. After incubation in the incubator for 8 or 24 h, 10 μL CCK-8 solution was added to each well for another 4 h. Next, the absorbance at 450 nm was measured using a microplate analyzer (Tecan Infinite). Each experiment was performed three times. The percentages of cell viability were calculated by (OD value of the Astral hydrogel group)/(OD value of the blank control) × 100%.

**Cell viability on gel surface.** Acridine orange/ethidium bromide (AO/EB, Solarbio) staining was used to distinguish living and dead cells. AO is permeable to both dead and living cells, while EB can penetrate dead cells and bind to their DNA and RNA. As a result, living cells will appear green and dead cells will appear yellow or orange. In the astral gel group, L929 cells (10⁵ cells/mL, 0.3 mL/well) were seeded on the surface of the astral gel, and an appropriate amount of culture medium was added to immerse the gel. In the control group, cells were seeded directly on the petri dishes with a culture medium. After 8 or 24 h incubation in 5% CO₂ incubator at 37 °C, the supernatant of the petri dishes was removed and washed twice with PBS. Dyeing solution was prepared by mixing AO (100 μg/mL), EB (100 μg/mL), and PBS buffer solution at the ratio of 1:1:100. About 500 μL staining solution was added to each well and incubated at room temperature without light for 10 min. The staining solution was removed and 100 μL PBS was used to wash off the unbound dye. The cells were observed under a fluorescence microscope and images were taken in random areas.

**Derivation of the minimal model.** We have discussed Eqs. 1 to 4 in the main text, while we shall focus on the expressions for $v_{IR}$ and $K_{poly}(\lambda, T)$ in this section. Consider two identical asters with a diameter $R$ stacked in the axial direction with a center-to-center distance $d$. We define a state of phantom interpenetration (conceptually analogous to phantom networks), in which the ribbons from one aster can freely pass their counterparts from the other aster yet they can still develop crosslinks by interlacement. In the phantom limit, the interpenetration region is the intersection of two perfect spheres with its height $h_P$ and volume $v_{IR}$ given by

$$h_P = R - d \tag{5}$$

$$v_{IR} = \pi \frac{h_P^2}{4}\left(R - \frac{h_P}{3}\right) \tag{6}$$

In the native state of a freshly prepared hydrogel in which two asters are moderately interpenetrated with $d = d_0$ ($\lambda_0 = 1$), the phantom limit is applicable because the ribbons are formed in situ and they can interlace without creating significant internal stress. Upon compression (or extension), the two asters are repositioned to $d = \lambda d_0$; the preexisting interlacement will resist interpenetration (or retraction) to some degree, modifying the height of the interpenetration region to $h_{NP}$ with the subscript NP denoting nonphantom. It is reasonable to expect a fraction of the global strain to be transmitted to the interpenetration region, giving.

$$\frac{h_{NP} - h_P}{h_P} = T \frac{\lambda - \lambda_0}{\lambda_0} \tag{7}$$

The prefactor $T$ is defined as strain transmittance from 0 (phantom interpenetration) to 1 (completely nonphantom interpenetration). Accordingly, the volume is now given by

$$v_{IR} = \pi \frac{h_{NP}}{4}\left(Rh_P - \frac{h_P^2}{2} + \frac{h_{NP}^2}{6}\right) \tag{8}$$

Notably, Eq. 8 reduces to Eq. 6 in the phantom limit when either $\lambda = 1$ or $T = 0$. Combining Eqs. 5, 7, and 8, and given two input parameters ($d_0$ and $T$), we are able to predict the dependence of normalized $v_{IR}$ on $\lambda$. As demonstrated in Supplementary Fig. 4, the effect of $d_0$ is significant, while the effect of $T$ is much less pronounced and

localized to the low-$\lambda$ region. Also, specific aster packing has an insignificant effect on the curves (Fig. 5b).

Next, we notice that the dependence of K on axial strain for networks of semiflexible polymers was obtained experimentally and by simulations[18]. We digitize the simulation curve (triangles in Fig. 2b of ref. [18]) with a polynomial function $K_{poly}$,

$$K_{poly}(x) = P_1 x^4 + P_2 x^3 + P_3 x^2 + P_4 x^1 + P_5 \tag{9}$$

where $x$ is the stretch ratio and the five coefficients are determined by fitting. In our case, $x$ is the stretch ratio local to the interpenetration region, reading

$$x \equiv \frac{h_{NP}}{h_P} = T\lambda - T + 1 \tag{10}$$

**Reporting Summary**. Further information on research design is available in the Nature Research Reporting Summary linked to this article.

## Data availability
The data that support the findings of this study are available in the main text or the Supplementary Information and are available from the corresponding author upon reasonable request.

## Code availability
The Matlab code to calculate the minimal model proposed in this paper is available from the corresponding author upon reasonable request.

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

## Acknowledgements
L.J. is in debt to Xixi Chen and Yuchao Li from Jinan University for their help on the optical fiber experiments. L.J. acknowledges support by National Natural Science Foundation of China (No. 21773092), Guangdong Natural Science Funds for Distinguished Young Scholar (No. 2018B030306011), The Recruitment Program of Guangdong (No.2016ZT06C322), Major Program of National Natural Science Foundation of China (51890871), and the Fundamental Research Funds for the Central Universities (No. 21617320).

## Author contributions
Q.X. and L.J. conceived the experiments. Q.X. synthesized the surfactant, discovered the asters, and carried out most of the experiments. Y.Z. and G.Y. performed the cell viability assays. T.W. and Y.C. performed the AFM measurements. All the authors analyzed the data and contributed to discussing the results and writing the paper.

## Competing interests
The authors declare no competing interests.
