## [Peer Review File · Nature Communications]

REVIEWER COMMENTS

Reviewer #1 (Remarks to the Author):

The manuscript entitled "Astral hydrogels mimic tissue mechanics by aster-aster interpenetration" by Qingqiao Xie, Gaojun Ye, and Lingxiang Jiang describes astral hydrogels that mimic mechanically tissue. This is uncommon as hydrogels are typically inert (as is the case for agarose and many others) or they exhibit mechanics opposite to those of tissues. The novel mechanics of the asters are compression-stiffening, extension-softening, and shear strain softening which occur due to the interpenetration ribbon network. The asters have a unique mechanism of aster-aster interpenetration that results in crosslinks between asters via entanglement of the ribbons. In order to test the mechanics, optical fibers were used instead of optical tweezers due to the size of the asters. The mechanics were tested multiaxially in order to mimic in vivo tissue environments. Historically hydrogels have been tested unilaterally which does not give an accurate estimate on how that gel compares to a tissue due to the fact that tissues are subjected to stress and strain from multiple angles. Upon using optical fibers to shear the asters it was observed that aster clusters can sustain greater strain when in a pre-compressed state than in the native state. However, the native state was still shown to stiffen and soften given axial strain. The effect of Aster gels on the mechanics of other gels once combined was also researched. On their own, agarose and PAAm exhibit a decreasing G/G_0 indicating that they do not mimic tissue mechanics. When asters were added into each of those gels, the G/G_0 of the gels increased resulting in moderate compression stiffening of the previously inert gels.

In summary, the astral gels stiffen in compression and soften in extension. The authors propose that during compression number of effective crosslinks increases in the interpenetration region, and the opposite in extension. The authors find that the interpenetrative nature plays a central role, in the sense that compression furthers interpenetration to stiffen the network and extension does the opposite, while the nonlinear elasticity of the semiflexible astral ribbons and network densification/dilution are secondary

The findings in this paper are novel and will be interest to the diverse audience of Nature Communication. The experiments are well thought out and executed. I recommend publication after the following minor comments are addressed.

A very nice paper to read and exciting findings.

Major and Minor Comments

- Please describe the temperature dependence and requirement for formation of this unique hydrogel. This will determine the handleability. If this is a limitation to its potential uses? Do you see these materials being using in vivo or for tissue engineering applications?
- Do the optical fibers affect the structural integrity of the hydrogels when moving the asters closer and farther away from each other?
- Do these "walls" built between the asters and the related strength of them remain after optical fibers are removed or are they only temporary?
 - o If so, how long does this strained/cluster state stay intact?
- What concentration of asters would be needed to turn an inert gel into a tissue-like gel? Or What amount of aster do you need to add to a simply hydrogel to get the effect? What is the concentration dependence of this effect.
- Have you performed any cytotoxicity studies? As these are multi-cationic systems – they maybe toxic?
- Can a smaller hydrophobic chain length be used for the surfactant instead of 16-2-16 to get the same aster formation? Might allow it to be heated to a lower temperature because of solubility?
- Do asters form at 30C (human skin temperature) or 37C (internal human temperature) or is it only at 25C? This will determine its potential use as a biomedical device/tissue mimetic.
- How large are the asters is diameter?
- What is the response rate after apply a stress for the hydrogel to return to its native state

Overall the text is well written and I suggest the following grammatic change.

Line 193 - To put the astral hydrogels in a comparative context, Please rewrite and do not use the phrase 'To put ...' In order to compare Or.... In a comparative contextOr....

Reviewer #2 (Remarks to the Author):

The manuscript describes the mechanical behavior of gemini astral structures under stress conditions. Due to the need of biomaterials that replicate the characteristics of soft tissues, the study is timely and valuable for biomaterials sciences.

However, there are a number of concerns that could improve the value of the paper.

- Reference 1 should be narrowed down to appropriate section of the book. References 2 and 3 too old. The first statement is overreferenced.

-In addition of the description of the images 2c-e, more explanation needed on the contribution of ee on the formed structures. Also, it is not clear whether 'interlacement' and 'entanglement' are distinct phenomena and in what context they are present.

-Description of figure 3 is good, however, interpretation is needed in the text with regards to the nature of the astral structures and composition of the gel. As these are relatively simple tests, the other ee values (0 and 0.5) should be tested to establish a structure-function relationship.

-While the experimental design for the real-time assessment of the aster-aster interaction by microscopy is valuable, assessing only one system limits the ability to draw generalizable conclusions.

-The argument at the end of ' Microscopic insights into aster-aster interpenetration' section is vague and needs substantiated.

-It is not clear why the experiments of incorporation of astral structures in inert gels is incorporated in the discussion section, rather than in the experimental section. Description and interpretation of figure 6 in the text is needed.

The rest of discussion is more amenable to a 'Conclusion' section. The discussion should be linked back to the introduction and outline how this study situates in the context of synthetic materials that mimic the shear, compression and extension behavior of biological tissues(with appropriate current references).

Reviewer #3 (Remarks to the Author):

In their submitted manuscript, Xie et al demonstrated synthesis of soft hydrogel biomaterials with a unique combination of tissue-like mechanical properties (shear-softening, compression-stiffening, and extension-softening) that appears to have not been achieved before. The gels are formed through interaction of synthetic asters (developed previously by the same group and reported in a 2019 Nature Communications manuscript – ref 23) in aqueous environments. Most interestingly, the authors observe compression stiffening of their gels, attributed to increased aster interpenetration and non-covalent association/crosslinking upon compression. The writing and visuals are very clear, and the methods described appropriately. Though the "master equation" is somewhat outside of my wheelhouse (and I leave other reviewers to comment on this analysis), the rheological characterizations and conclusions drawn appear appropriate. Overall, I found this approach to be unique and likely of interest to the Nature Communications readership. I anticipate recommending that the article be accepted for publication after addressing the following points:

- Based on images shown in Fig. 2, it appears that there is fairly substantial heterogeneity in both aster size and their in-gel packing. While rheological analysis is useful at measuring bulk properties of these gels, techniques to assess local mechanical variation should also be performed and included. Atomic force microscopy is one such technique offering spatial resolution of the

samples that would add a lot to the current manuscript.

- How does the rate of compression-stiffening compare with that of native tissue?
- At line 104, the authors note that the slope is invariant to ϵ , which appears counterintuitive. Was this expected? Additional justification should be included in the text.
- Despite comments listed at the bottom of Page 9, it is not fully clear in what way these materials could be used. Can cells be encapsulated within these gels?
- How sensitive are the gels to different buffer conditions? Is erosion observed?

Reviewer #4 (Remarks to the Author):

The present study revealed that the hierarchical self-assembled aster gel system exhibits the living-tissue-specific mechanics of compression stiffening and extension softening, which has a potential advantage to design a new class of biomaterials with tissue-mimic mechanics. Regulating axial strains for the gels, shear viscoelasticity response was systematically measured, and micro compression/shear-loading experiments were performed using optical fibers. They demonstrated compression stiffening/extension softening of the aster gels, succeeded in obtaining a strain-moduli master curve, and discussed a simple successful mechanics model that reproduces well the behavior of the master curve. Though the data are solid enough, the logic is almost clear, and total content is very interesting, there is some concern to be clarified.

The central findings are in the strain-moduli master curve and in the proposed model that explains it. According to the result from the model in Fig.5b, interpenetration volume between aster gels was found as the strongest contributor for generating the master curve irrespective that it is phantom or nonphantom character. This means that the essential cause for the aster gel system to show the compression stiffening/extension softening is in the interpenetration region and crosslinking density there. On the other hand, as we consider mechanics of living epithelial tissue, there are no such an interpenetration volume between cell-cell space, because they form tight adherent junction each other. In the epithelial tissue, tissue are constructed with tight cell-cell contact, and there are no interpenetration elements in the system as depicted in this study, though a cell has aster-like architecture of microtubule cytoskeleton inside the cell membrane. They cited the references of 19-21 in this article as a representative example of the tissue with compression stiffening/extension softening, but which are limited to such as brain tissue and mesenchymal tissue with low cell density and rich mesenchymal extracellular matrix like aster-mimic conditions. Thus, it may somewhat be an overstatement to address the general mechanics of living tissue without mentioning the mechanics of epithelial tissue.

The author should mention whether the compression stiffening/extension softening also occurs in epithelial tissues, and if it is valid, they should discuss why the compression stiffening/extension softening can occur even in the absence of interpenetration volume between the cells as mechanics element. These issues are at the heart of the paper and will need to be clarified if they want to establish the aster-gel system as the general mimic of living tissue mechanics.

Revision Requested for Manuscript NCOMMS-20-38198

Astral hydrogels mimic tissue mechanics by aster-aster interpenetration

Response to Reviewer # 1

We thank the reviewer for the insightful comments and the recommendation of publication after revision. We have done our best, in the revised manuscript, to address the reviewer's concerns.

1. Concerns about the handleability and in vivo applications of the astral hydrogels.

“Please describe the temperature dependence and requirement for formation of this unique hydrogel. This will determine the handleability. If this is a limitation to its potential uses? Do you see these materials being using in vivo or for tissue engineering applications?”

Preparation of the astral hydrogels is as simple as that of agarose hydrogel. We weighed desired amount of surfactant powder in water, dissolved the powder at 80 °C, and then incubated the sample at 25 °C for a few hours for astral networks to be fully developed. The critical temperature of aster formation is around 45 °C, and gel formation is not sensitive to cooling rate (we tried controlled cooling from 1 to 10 °C/min or ambient cooling). Overall, the hydrogel formation is simple and robust, posing no limitation towards its potential uses. Part of this paragraph is now added to the methods section.

These materials, as far as we know, have not been used in vivo or in tissue engineering. Since the 16-2-16 molecule is essentially a cationic surfactant, we suspect certain toxicity of it. We anticipate the deposition of a layer of silica or other inert materials onto the surface of asters to screen the 16-2-16 molecules, which may minimize its toxicity and enhance its stability. Part of this paragraph is now added to the discussion section.

2. Concerns about the structural integrity during optical fiber manipulation.

“Do the optical fibers affect the structural integrity of the hydrogels when moving the asters closer and farther away from each other?”

The integrity of individual asters is largely preserved during manipulation by optical fibers (see also Fig. 5e and Movie S5 of our previous paper on the synthetic asters, Nat Comm 2019, 10, 4954). The integrity of the hydrogels is hard to assess, because, in the micromanipulation experiments, we took a small piece of a hydrogel and dispersed it in excess water so that we can manipulate small clusters or individual asters. Otherwise, the astral gel is an intact and tight block of asters. This detail is now added to the methods section.

3. Concerns about the life time of aster-aster junctions.

“Do these “walls” built between the asters and the related strength of them remain after optical fibers are removed or are they only temporary? If so, how long does this strained/cluster state stay intact?”

When two separate asters are pushed against each other by optical fibers, an aster-aster junction (visible ‘wall’) develops. When the optical fiber is removed, the asters will ‘bounce back’ a bit (in another word, the interpenetration will retreat a bit) possibly due to aster elasticity, but the junction is largely preserved permanently. Two asters will detach only if they are pulled away by optical fibers (see Movie S9 in Nat Comm 2019, 10, 4954). Therefore, a strained cluster will remain intact unless it is forcefully dismembered by optical fibers.

4. Concerns about the concentration dependence of addition of asters to inert gels.

“What concentration of asters would be needed to turn an inert gel into a tissue-like gel? Or What amount of aster do you need to add to a simply hydrogel to get the effect? What is the concentration dependence of this effect.”

There is indeed a critical concentration above which the added asters can render the inert gel compression-stiffening. For PAAm and agarose gels, this critical concentration of asters is about 1 to 2 mg/ml. Below this concentration, the asters are not populated enough to form a connective network. Above this concentration, compression-stiffening is always observed and the overall elasticity (G') increases with aster concentration. The slope of compression-stiffening (‘responsiveness’ defined in Fig. 6a) is weakly dependent on aster concentration. Part of this discussion is now added to the revised manuscript.

5. Concerns about toxicity and chain length of the 16-2-16 molecules.

“Have you performed any cytotoxicity studies? As these are multi-cationic systems – they maybe toxic?”

Indeed, we expect a certain level of cytotoxicity of the 16-2-16 molecules because of its cationic headgroups. Although we have not performed cytotoxicity experiments, this surfactant was reported to kill bacterial and HeLa cells. For example, viability of HeLa cells in the presence of 0.5 mg/ml 12-2-12 for 4 hours is around 40% (Colloids and Surfaces B: Biointerfaces 179 (2019) 437–444). The cytotoxicity of 16-2-16 is anticipated to be lower than that of 12-2-12 due to its significantly lower solubility and CMC.

“Can a smaller hydrophobic chain length be used for the surfactant instead of 16-2-16 to get the same aster formation? Might allow it to be heated to a lower temperature because of solubility?”

Chain length of the gemini surfactant has been systematically studied by Oda et. al (Chem. Commun., 1997, 2105-2106), where they found that a shorter chain length (for example, 12-2-12) leads to the formation of wormlike micelles instead of rigid fibers or ribbons. Therefore, we do not anticipate aster formation for shorter hydrophobic chains.

6. Concerns about formation temperature.

“Do asters form at 30C (human skin temperature) or 37C (internal human temperature) or is it only at 25C? This will determine its potential use as a biomedical device/tissue mimetic.”

The critical gelation temperature is ~ 45 °C (it is also the Kraft point of 16-2-16). So the asters can form below this temperature, enabling its potential use in human skin temperature and internal human temperature.

7. Concerns about aster diameter and stress response rate.

“How large are the asters is diameter? What is the response rate after apply a stress for the hydrogel to return to its native state”

The diameter ranges from 20 to 120 microns, its average is around 50 micron for 1 mg/ml hydrogels. In practice, we apply compression to a gel sample and wait a few minutes for the sample to reach a static state; then its stiffening or softening response is measured. The kinetics in response is not specifically measured, but we do notice that the stiffening or softening is finished within seconds to tens of seconds. When the compression is removed, the gel can only partially recover to its native state within minutes. We plan to carry out detailed study of time dependence in a future work.

8. A grammatic change.

“Line 193 - To put the astral hydrogels in a comparative context, Please rewrite and do not use the phrase ‘To put ...’ In order to compare Or.... In a comparative contextOr....”

We thank the reviewer for this suggestion and have changed it into “In a comparative context...”.

Response to Reviewer # 2

We thank the reviewer for the critical reviewing and constructive suggestions. We have done our best to address the reviewer's concerns in the revised manuscript.

1. Concerns about the references.

“Reference 1 should be narrowed down to appropriate section of the book. References 2 and 3 too old. The first statement is overreferenced.”

A specific chapter of ref 1 is now cited. Ref 2 and 3 are removed, so the first statement is now supported by 2 refs.

2. Concerns about effect of ee and difference between ‘interlacement’ and ‘entanglement’.

“In addition of the description of the images 2c-e, more explanation needed on the contribution of ee on the formed structures. Also, it is not clear whether ‘interlacement’ and ‘entanglement’ are distinct phenomena and in what context they are present.”

The ee has a dramatic effect on helicity of the aster ribbons (Supplementary Fig. 2 in Nat Comm 2019, 10, 4954), but its effect on the overall astral geometry is limited (Fig. 2c-2e of this paper). In particular, visual inspection of the microscopic images suggests that the asters are of similar sizes and similar ribbon densities for different ee values. We meant ‘interlacement’ and ‘entanglement’ for synonyms in this paper. We have now dropped all the ‘entanglement’ and kept ‘interlacement’ to avoid ambiguity.

3. Concerns about structure-function relationship.

“Description of figure 3 is good, however, interpretation is needed in the text with regards to the nature of the astral structures and composition of the gel. As these are relatively simple tests, the other ee values (0 and 0.5) should be tested to establish a structure-function relationship.”

We have indeed tested samples with different ee values (but not all experimental data are presented). Samples with ee = 0 or 0.5 behavior similarly to the ee = 0.33 sample (curves similar to those in Fig. 3a and 3b). The effect of ee is demonstrated in Fig. 3e, where one can notice that the slope of the compression-stiffening response is largely invariant to ee. In the “A minimal model to describe the axial mechanics of astral gels” section, we reason that the compression-stiffening is mostly affected by the overall astral geometry (for example, ribbon density) rather than any molecular specifics or ribbon

properties (such as persistence length or helicity). Therefore, the compression-stiffening slope is invariant to ϵ as we observed. Part of this discussion is now added to the revision.

4. Concerns about the generalizable conclusions of aster-aster interaction.

“While the experimental design for the real-time assessment of the aster-aster interaction by microscopy is valuable, assessing only one system limits the ability to draw generalizable conclusions.”

We agree that assessment in one system is limited. Actually, we have performed multiple microscopic tests (optical fiber manipulation) for aster-aster interactions and aster cluster compression for different 16-2-16 concentrations (≥ 3 times for each concentrations), and we obtained similar observations. The relevant conclusions are therefore generic for the current aster system. We have now clarified this in the revision.

5. Concerns about the specific arguments in the end of the microscopic section.

“The argument at the end of ‘Microscopic insights into aster-aster interpenetration’ section is vague and needs substantiated.”

In the end of the mentioned ‘microscopic’ section, we made the arguments that the moderate aster interpenetration in the native state can mechanically support the gel and that axial strain can advance or withdraw such interpenetration, effectively stiffening or softening the gel. We agree that these are comprehensive claims and they need to be further supported and substantiated. Indeed, these claims are motivated from the microscopic experiments and substantiated mainly by the mechanical model and the fitted master curve in the next section. In particular, the interpenetration region is defined, and expansion and retreat of this interpenetration region is related to the stiffening and softening. These claims are also supported by the matchiness between our model and the rheological curve (Fig. 5a-5b). The ambiguity in the end of the ‘microscopic’ section is now clarified.

6. Concerns about contents and arrangement in the discussion section.

“It is not clear why the experiments of incorporation of astral structures in inert gels is incorporated in the discussion section, rather than in the experimental section. Description and interpretation of figure 6 in the text is needed. The rest of discussion is more amenable to a ‘Conclusion’ section. The discussion should be linked back to the introduction and outline how this study situates in the context of synthetic materials that mimic the shear, compression and extension behavior of biological tissues (with appropriate current references).”

We thank the reviewer for this suggestion. We have now moved ‘the incorporation of asters to inert gels’ to results section. We agree that part of previous discussion is more like a conclusion section, but the Nat Commun format discourage a specific “conclusion” section. The discussion section is now substantially enriched by contents on the comparison between tissues and astral gels and on the potential applications and concerns on astral gels.

Response to Reviewer # 3

We thank the reviewer for the positive comments and insightful suggestions. The manuscript is now substantially revised in response to the reviewer's suggestions.

1. Suggestion of using AFM to obtain spatially resolved properties.

“Based on images shown in Fig. 2, it appears that there is fairly substantial heterogeneity in both aster size and their in-gel packing. While rheological analysis is useful at measuring bulk properties of these gels, techniques to assess local mechanical variation should also be performed and included. Atomic force microscopy is one such technique offering spatial resolution of the samples that would add a lot to the current manuscript.”

Indeed, there are quite some inhomogeneity in terms of sizes and packing of asters, which could affect local mechanical properties. We appreciate the reviewer's suggestion to perform AFM measurements to spatially resolve the local mechanics. Actually, we have previously done AFM experiments on single asters and observed elasticity dropping from aster center to aster periphery (Fig. 5a-5b, Nat Comm 2019, 10, 4954). We have also tried to employ AFM for the current study but encountered certain difficulties.

First, the AFM cantilever is perfectly suited to vertically deform the asters, but it is hard to shear the asters horizontally. The AFM technique is thus difficult to probe the unique mechanical property of focus in this paper (the biaxial response: compression in the vertical direction and stiffening in the horizontal direction).

Second, the AFM technique is usually applicable for a thin layer of materials. In our case, a single layer of asters (~50 micron in diameter) is thick enough for AFM measurements. We have done preliminary experiments to measure the elasticity for a single layer of asters and indeed observed significant spatial variation. But we reason that a single layer of asters is substantially different from the bulk of an aster hydrogel where the asters are adjacent to their neighbors in 3 dimensions.

Overall, we agree that it is highly valuable to measure the local properties and to obtain the site-to-site variance, but our preliminary attempts indicate that AFM is not a very suitable means in this case. We are currently looking for other possible means (such as microrheology with imbedded probe particles) to resolve the spatially dependent mechanics of astral hydrogels.

2. Concerns about rate of compression-stiffening.

“How does the rate of compression-stiffening compare with that of native tissue?”

If the ‘rate’ of compression-stiffening was meant for how fast the gel is compressed or how fast the gel responds by stiffening itself, we would like to clarify that the experiments were mainly carried out in a static manner. In practice, we apply compression to a gel sample and wait a few minutes for the sample to reach a static state; then its stiffening or softening response is measured. The kinetics in response is not specifically measured, but we do notice that the stiffening or softening is finished within seconds to tens of seconds. When the compression is removed, the gel can only partially recover to its native state within minutes. We plan to carry out detailed study of time dependence in a future work.

If the rate of compression-stiffening was meant for how strong the stiffening response is, then we did define it as ‘responsiveness’ and compared it (around 5) with that tissues (from 4 to 18) in Fig 6a. Part of this discussion is now added to the revised manuscript.

3. Concerns about effect of ee on the compression-stiffening slope.

“At line 104, the authors note that the slope is invariant to ee, which appears counterintuitive. Was this expected? Additional justification should be included in the text.”

Indeed, it may appear a bit surprising to see that the compression-stiffening slope is insensitive to ee, and a word of explanation is well deserved. The ee has a dramatic effect on helicity of the aster ribbons (Supplementary Fig. 2 in Nat Comm 2019, 10, 4954), but its effect on the overall astral geometry is limited (Fig. 2c-2e of this paper). In particular, visual inspection of the microscopic images suggests that the asters are of similar sizes and similar ribbon densities for different ee values. In the “A minimal model to describe the axial mechanics of astral gels” section, we showed that the compression-stiffening is mostly affected by the overall astral geometry (for example, ribbon density) rather than any molecular specifics or ribbon properties (such as persistence length or helicity). Therefore, the compression-stiffening slope is invariant to ee as we observed. This discussion is now added to the revision.

4. Concerns about the applicable situations of the aster hydrogels.

“Despite comments listed at the bottom of Page 9, it is not fully clear in what way these materials could be used. Can cells be encapsulated within these gels?”

Although the space between aster fibers (on the order of 100 nm) is too small to accommodate cells, we envision the asters to be used as extracellular matrix to connect different cells. For example, cells can be mixed with the 16-2-16 monomers and encapsulated in the astral gel during gel formation. One issue is the potential cytotoxicity of 16-2-16, because this molecule is essentially a multiple-cationic surfactant. We anticipate the deposition of a layer of silica or other inert materials onto the surface of asters to screen

the 16-2-16 molecules, which may minimize its toxicity and enhance its stability. Part of this paragraph is now added to the discussion section.

5. Concerns about aster stability in different buffer conditions.

“How sensitive are the gels to different buffer conditions? Is erosion observed?”

The asters are quite stable against a range of pH (4 to 12) and moderate ionic concentrations (up to 0.1 M NaCl). Too much acid will affect ionization of the counterion (tartrate), and a high ionic concentration will screen the electrostatic interaction between the 16-2-16 headgroups and tartrate counterion. In addition, the asters are stable against reduction/oxidization agents. The astral gels are therefore insensitive to common buffer conditions, and no erosion was observed. Part of this paragraph is now added to the discussion section.

Response to Reviewer # 4

We thank the reviewer for the positive feedbacks and insightful concerns on epithelial tissues. A part of the discussion section in the revised manuscript is now dedicated to compare the current astral gels to epithelial and other tissues.

1. Concerns about the difference between epithelial and other kind of tissues, and their relation to the current astral gels.

“In the epithelial tissue, tissue are constructed with tight cell-cell contact, and there are no interpenetration elements in the system as depicted in this study...

...it may somewhat be an overstatement to address the general mechanics of living tissue without mentioning the mechanics of epithelial tissue.

The author should mention whether the compression stiffening/extension softening also occurs in epithelial tissues, and if it is valid, they should discuss why the compression stiffening/extension softening can occur even in the absence of interpenetration volume between the cells as mechanics element. These issues are at the heart of the paper and will need to be clarified if they want to establish the aster-gel system as the general mimic of living tissue mechanics.”

First, we agree that the epithelial tissues are quite different from other soft tissues such as fat or brains in the sense that the cells are tightly packed with minimal extracellular matrix. Current research on compression-stiffening or compression-softening is limited to connective tissues (like fat) or nerve tissues (like brains). There is no research yet, as far as we know, to indicate how the epithelial or muscle tissues would respond mechanically in the shear direction under axial compression. Given the significant difference between different tissue types, they may well behave differently. Therefore, we do not claim that the compression-stiffening is generic for all soft tissue types, but this feature is indeed common among many connective and nerve tissues (such as adipose, kidney, liver, lung, brain, and spinal cord tissues). We thank the reviewer for pointing this out, and have specified this in the revised manuscript.

Second, we observed compression-stiffening in astral gels and attributed this response to the interpenetration region. This phenomenon is similar to the compression-stiffening observed in the above-mentioned tissues, but the mechanism is likely to be different. Recently, Janmey et. al. studied the mechanism for tissues' compression-stiffening behavior and proposed a model for it (Nature 2019, 573, 96-101). The extracellular matrix is not interpenetrative (like common gels). They considered that the embedded cells can convert the global compression into local stretch for the semiflexible filaments (extracellular matrix) adjacent to the cells. If the inclusions are densely populated, the local effect of stretch-stiffening prevails such that the tissues behave in a compression-stiffening manner. Therefore, we

would like to state that the current mechanism on the basis of aster-aster interpenetration is fundamentally different, and that it may provide an alternative yet effective route towards compression-stiffening mechanics. Part of this discussion is emphasized in the discussion section.

REVIEWER COMMENTS

Reviewer #1 (Remarks to the Author):

The authors have adequately addressed my comments and the manuscript is improved. I recommend publication. Nice study.

Reviewer #2 (Remarks to the Author):

Most of the revisions seem acceptable. However, the changes are not highlighted, therefore is not possible to distinguish the new sections.

Point 3 of rebuttal is not addressed; the authors describe that other ϵ 0 to 0.5 behave similarly, data is necessary, at least as supplementary data. Establishing structure-activity relationship is possible only if a series of conditions are examined and presented.

Point 4 is partially addressed. While many microscopic examinations were performed, it was still on a single surfactant 16-2-16 and a single ϵ .

Some grammatical corrections are needed.

It is suggested that all editing to be highlighted.

Reviewer #3 (Remarks to the Author):

Though the authors have responded satisfactorily to some of my comments, I still feel that the following points need to be more fully addressed prior to publication:

- The manuscript should include efforts to assess spatial variation in network mechanics. The authors state in their response that AFM has proven challenging, but well-established alternative techniques including microrheology could be employed.
- While these are very interesting materials, the use-case for these gels remains underwhelming in the revised document. The authors should include simple cell encapsulation studies and demonstrate viability. Without these, much of the motivation of these being more relevant synthetic tissues seems unjustified.

Reviewer #4 (Remarks to the Author):

My concern on the general statement for compression-stiffening including epithelial tissues has been discussed appropriately in the revised version.

Revision Requested for Manuscript NCOMMS-20-38198A

Astral hydrogels mimic tissue mechanics by aster-aster interpenetration

Please note that 3 authors (Yuandi Zhuang, Tiankuo Wang, and Yi Cao) have been added to the author list for their contributions in the new experiments of this revision.

Response to Reviewer # 1 and Reviewer # 4

Reviewers # 1 and # 4 were satisfied with the revision and have no further questions. We thank them for their approval.

Response to Reviewer # 2

1. Suggestion to highlight the changes.

“Most of the revisions seem acceptable. However, the changes are not highlighted, therefore is not possible to distinguish the new sections.”

In the previous revision, two versions of manuscript were submitted: one is a “clean” version (main file), the other is a “change-tracked” version (supplied as an additional file). This time, we have submitted the tracked version as the main file and highlighted all the changes as the reviewer suggested.

2. Concerns about the impact of different ee .

“Point 3 of rebuttal is not addressed; the authors describe that other ee 0 to 0.5 behave similarly, data is necessary, at least as supplementary data.”

We now present the results of different ee here and in the supporting information. In frequency sweep and shear strain sweep experiments, samples with $ee = 0$ or 0.5 behavior similarly to the $ee = 0.33$ sample (concentration = 1.5 wt %, panel **a** and **b** below). Panel **c** is a copy of Fig. 3e, showing that the slope of the compression-stiffening response is largely invariant to ee . In the “A minimal model to describe the axial mechanics of astral gels” section, we reason that the compression-stiffening is mostly affected by the overall astral geometry (for example, ribbon density) rather than any molecular specifics or ribbon properties (such as persistence length or helicity). Therefore, the compression-stiffening slope is invariant to ee as we observed. This discussion is added to the supporting information.

3. Suggestion to perform more microscopic examinations.

“Point 4 is partially addressed. While many microscopic examinations were performed, it was still on a single surfactant 16-2-16 and a single *ee*.”

We have synthesized two additional surfactants, 12-2-12 and 14-2-14, but they cannot form asters. So we still focused on 16-2-16 and performed extra optical fiber experiments on samples with different *ee* values. D_{10}/D_{00} and D_{11}/D_{10} reflects microscopic yield points in uncompressed and pre-compressed states, respectively. Notably, D_{11}/D_{10} is larger than D_{10}/D_{00} in a similar fashion for *ee* = 0, 0.33, and 0.5. This result suggests that the compression-stiffening response is microscopically similar for samples with different *ee* values. In addition, two typical aster manipulation movies for *ee* = 0 and 0.5 samples are also provided.

Movie S2

Movie S3

4. “Some grammatical corrections are needed.”

We have done our best to scrutinize this revision and repolish it with the aid of a native English speaker.

Response to Reviewer # 3

We thank the reviewer for the positive comments and insightful suggestions. The manuscript is now substantially revised in response to the reviewer's suggestions.

1. Suggestion to assess spatial variation of the network.

“The manuscript should include efforts to assess spatial variation in network mechanics. The authors state in their response that AFM has proven challenging, but well-established alternative techniques including microrheology could be employed.”

In this revision, we have performed spatially resolved modulus measurements by AFM as the reviewer suggested.

Experimentally, the Young's modulus is mapped out for a layer of astral gel by an AFM (JPK Nanowizard II, JPK Instruments, Berlin, Germany) at room temperature. The common sharp tip is replaced by a 7- μm silica sphere to ensure smooth compression on loose astral ribbons; to do so, the silica sphere was attached onto the silicon nitride tip (type MLCT, Bruker Company) by epoxy glue. A cantilever (spring constant in the range of 0.03-0.07 N m^{-1}) was positioned on top of a thin layer of astral gel. Multiple (200 to 300) intention-retraction curves were retrieved for a single position; identical measurements were performed on a 5×4 lattice. Each approaching curve is fitted by the Hertz model to give a Young's modulus, E , and an averaged value is obtained for different positions on the gel layer.

As shown in the figure below (panel a), the gel surface height fluctuates on the order of 500 nm. Regularly separated positions on a lattice are labeled as 0 to 19, the moduli of which are summarized in panel b and c. Notably, the spatial variation is significantly large from 0.3 to 50 kPa, suggesting that inhomogeneity in aster sizes and packing could affect local mechanical properties. There is no obvious spatial correlation in modulus in panel c. The overall mean E is 15 kPa, in agreement with $G' = 6$ kPa measured by rheometer ($E = 3G'$ assuming a Poisson's ratio of 0.5).

We would like to add a few words of explanation on the selection of spatially resolved mechanics methods. While AFM and microrheology techniques are routinely used to probe local mechanics, they are, as far as we know, still limited to the uniaxial responses. It is rather difficult to locally measure biaxial mechanics (e.g., compression in the vertical direction and stiffening in the horizontal direction) by both methods. For example, the AFM cantilever is perfectly suited to vertically deform the asters, but it is hard to shear the asters horizontally. Therefore, the spatial variation in Young's modulus as we measured in this section cannot imply the variation of compression-stiffening response.

2. Suggestions to perform cell viability experiments.

“While these are very interesting materials, the use-case for these gels remains underwhelming in the revised document. The authors should include simple cell encapsulation studies and demonstrate viability. Without these, much of the motivation of these being more relevant synthetic tissues seems unjustified.

We appreciate the reviewer's suggestion to quantify cell viability and have performed 2 assays in the revision. 1) The CCK8 assay that uses a water-soluble tetrazolium salt to quantify the number of live cells

by producing an orange formazan dye upon bio-reduction in the presence of an electron carrier. Specifically, an extracting solution of the astral gel is prepared and diluted by DMEM (dulbecco's modified eagle medium) to different extend; fibroblast cells (L929) are then incubated in these mixed solutions for 8 or 24 hours; finally, CCK8 reagents are applied and absorbance at 460 nm recorded to infer cell viability. Our results indicate that the constitute(s) extracted from the astral gel are of certain cytotoxicity that kills ~ 40 % cells in 24 hours.

2) The live and dead assay that employs acridine orange and ethidium bromide to label the live cells as green and the dead as green and red in a fluorescent microscope. L929 cells are incubated on the surface of astral gels for 8 or 24 hours, after which they are treated with the live/dead reagents and observed. We can see that the cells are alive in the control group but all dead on the astral gels. The fluorescent molecules seem to dye the gel fibers as well.

We did not conduct encapsulation experiments since embedment in the gels would be even harder for the cells to survive. From the above two experiments, we can conclude that the 16-2-16 molecule is

quite cytotoxic, similar to other dimeric cationic surfactants. Although the current astral hydrogels themselves are not suitable for cell culture or encapsulation, we argue that the key finding of this work lies on the interpretative nature of the astral gel beyond its molecular details. If one can replace 16-2-16 molecule with a cytocompatible molecule that forms asters, it is then possible to make cytocompatible astral gels. Alternatively, it is possible to deposit of a layer of silica or other inert materials onto the surface of asters to screen the 16-2-16 molecules, minimizing its toxicity and enhancing its stability.

REVIEWERS' COMMENTS

Reviewer #2 (Remarks to the Author):

The additional queries were addressed.

Reviewer #3 (Remarks to the Author):

I appreciate the authors' willingness to conduct and report AFM & cell viability studies, even if the results may be perceived as less than ideal. While I certainly wish that the gels were not cytotoxic and that 2D and 3D cell studies could be possible, I do appreciate the authors' material findings even without this ability. Though I still see this work as potentially interesting but without a current application, I can support publication of this work after the following points are addressed.

- Experimental details for studies for Supplementary Figures 6 and 7 are severely lacking. How many cells were seeded on each surface? What are "control gels"?
- Supplementary Figure 6 needs clarification as to what the error bars represent.
- Supplementary Figure 7 needs scale bars.

Revision Requested for Manuscript NCOMMS-20-38198B

Astral hydrogels mimic tissue mechanics by aster-aster interpenetration

Response to Reviewer # 2

Reviewers # 2 was satisfied with the revision and have no further questions.

Response to Reviewer # 3

1. *Experimental details for studies for Supplementary Figures 6 and 7 are severely lacking. How many cells were seeded on each surface? What are “control gels”?*

10⁵ cells were seeded on gel surface. In the control group, cells were seeded directly on the petri dishes with culture medium. A detailed description for the experiments is now added to the Methods section and pasted below.

Cell viability in the gel extract. The cell line, American ATCC mouse fibroblast L929, was used in this study. The cell culture medium consisted of 10% bovine serum albumin (Gibco), 1% penicillin-streptomycin (Gibco) and DMEM (Gibco). Cells and sterilized dishes were cultured in 5% CO₂ cell incubator at 37 °C. Cell counting kit 8 (CCK-8, Dojindo) assay was used for cytotoxicity evaluation of extracting solutions from the astral gels. First, L929 cells (5×10³ cells/mL) in the growth phase were inoculated in a 96-well plate (100 μL/well), and then the cells were cultured in a mixed solution of extract and culture medium. The volume fraction of the extract was 0%, 12.5%, 25%, 50%, and 100%, among which 0% was taken as the blank group. After incubation in the incubator for 8 or 24 hours, 10 μL CCK-8 solution was added to each well for another 4 hours. Next, the absorbance at 450 nm was measured using a microplate analyzer (Tecan Infinite). Each experiment was performed three times. The percentages of cell viability were calculated by (OD value of the Astral hydrogel group) / (OD value of the blank control) × 100%.

Cell viability on gel surface. Acridine orange/ethidium bromide (AO/EB, Solarbio) staining was used to distinguish living or dead cells. AO is permeable to both dead and living cells, while EB can penetrate dead cells and bind to their DNA and RNA. As a result, living cells will appear green and dead cells will appear yellow or orange. In the astral gel group, L929 cells (10⁵ cells/mL, 0.3 mL/well) were seeded on the surface of astral gel, and appropriate amount of culture medium was added to immerse the gel. In the control group, cells were seeded directly on the petri dishes with culture medium. After 8h or 24h incubation in 5% CO₂ incubator at 37°C, the supernatant of the petri dishes was removed and washed

twice with PBS. Dyeing solution was prepared by mixing AO (100 µg/mL), EB (100 µg/mL) and PBS buffer solution at the ratio of 1:1:100. 500 µL staining solution was added to each well and incubated at room temperature without light for 10 min. The staining solution was removed and 100 µL PBS was used to wash off the unbound dye. The cells were observed under a fluorescence microscope and images were taken in random areas.

2. Supplementary Figure 6 needs clarification as to what the error bars represent.

Error bars represent Standard Deviations (n=3). This statement is added to the caption.

3. Supplementary Figure 7 needs scale bars.

Scale bars are added now.